# Research Productivity among Filipino Neurologists Associated with Socioeconomic, Healthcare, and Disease Burden Factors: A Bibliometric Analysis

**DOI:** 10.3390/ijerph192315630

**Published:** 2022-11-24

**Authors:** Almira Doreen Abigail O. Apor, Roland Dominic G. Jamora

**Affiliations:** 1Department of Neurosciences, College of Medicine and Philippine General Hospital, University of the Philippines Manila, Manila 1000, Philippines; 2Institute for Neurosciences, St. Luke’s Medical Center Global City, Taguig City 1634, Philippines

**Keywords:** bibliometrics, Philippine neurology, Philippines, research productivity, neurology

## Abstract

Philippine research productivity in neurology has not been fully characterized. We investigated the research output of adult and child neurologists in the Philippines and correlated this to the Philippine socioeconomic and healthcare indices among different regions. We used electronic databases to retrieve studies published by Filipino neurologists using the 2022 Philippine Neurological Association website as reference. We included all studies published until December 2021. Official government region-specific socioeconomic indices were used. Correlational analysis was completed on bibliometric indices and collected data. We retrieved 746 articles from 274 of 526 Filipino neurologists which were published in 245 publications over 45 years with 12,409 citations. The National Capital Region (NCR) had the most publications (*n* = 662, 88.7%) and citations (*n* = 10,377, 83.6%). Research productivity was positively correlated with population, gross domestic product (GDP), health expenditure, number of healthcare establishments, neurologists, and research personnel. The Philippine research landscape is dominated by articles of neurologists belonging to institutions in the NCR, which has the greatest number of neurologists, training institutions, and highest GDP. There is a need to address the disparity seen in other regions to bridge gaps in healthcare, health human resources, and health information through research.

## 1. Introduction

The Philippines is an archipelago comprised of 17 administrative regions in 3 major islands (Luzon, Visayas, Mindanao) consisting of various topographies, which gives rise to several geographically isolated and disadvantaged areas. Governance fragmented into local government units has caused stark socioeconomic disparity and health inequity in that the public sector providing for the underserved is under-resourced whereas the private sector catering to the wealthy is over-resourced [1].

In Southeast Asia, the Philippines has shown the greatest increase in disability-adjusted life-years (DALYs) for neurological diseases over the last two decades [2]. A comparison of DALYs for neurological diseases between 1990 and 2016 showed an 87% increase, which highlights a growing demand for neurologists in the country.

The Philippine Neurological Association (PNA), formed in 1972, is the local authority for the specialty which accredits neurology training programs for both adult and pediatric neurology and board-certifies graduates of these programs. At present, there are 11 accredited training programs for adult neurology residency and 3 for pediatric neurology fellowship. The official PNA website lists five hundred adult and pediatric neurologists in the country [3]. The PNA strives for excellence in the delivery of neurological care, education, and research [3]. The role of the neurologist as a clinician is well-established, however, the role of a researcher remains to be elucidated.

In 2007, BRAIN Inc. compiled neuroscience research studies from 1985 to 2006 in the country in a compendium that included 610 abstracts, including 102 studies that have already been published locally [4]. This, however, was limited in that it only included abstracts, the majority of the submissions were unpublished, and the remaining published studies were in local journals with no automated citation counter.

Research productivity is defined as the number of peer-reviewed articles published and the frequency of citations [5]. Previous studies exploring research productivity for various neurologic diseases have found research disparity between low–middle income countries (LMIC) in comparison to higher-income countries [6]. In terms of research output in Southeast Asia, the Philippines ranks behind Singapore, Malaysia, and Thailand for studies on central nervous system infection, dementia, epilepsy, headache, motor neuron diseases, movement disorders, multiple sclerosis, and neuromyelitis optica, neuro-oncology, and stroke [6,7,8,9,10,11,12,13]. The exception is noted in the case of publications on dystonia, in which the Philippines ranked first [11]. Studies exploring treatment gaps for various neurological diseases in the country have shown a want of health information in the form of research. These studies cite a lack of local epidemiologic data and local experience with diagnosis or treatment for autoimmune encephalitis, bacterial meningitis, brain tumors, epilepsy, multiple sclerosis, Parkinson’s disease, and even stroke [14,15,16,17,18,19]. By further evaluating research performance, we can identify systemic disparities and guide appropriate solutions and direction for further study.

Thus, this study investigated the research output in terms of publications and citations of Filipino adult and child neurologists in the country and the correlation of these bibliometrics to socioeconomic and healthcare indices in the Philippines.

## 2. Materials and Methods

International (Scopus, PUBMED, Google Scholar) and local (Health Research and Development Information Network) electronic medical databases were used to retrieve studies published by Filipino neurologists using the PNA official website as reference. An initial total of 1310 publications were retrieved and extracted from the electronic databases using the following criteria: (a) authored by a Filipino adult or child neurologist (as indexed on the official PNA website as of June 2022); (b) published completed studies until December 2021 of any study design; (c) book chapters in published text; (d) available in full text; (e) published under at least one institution in the Philippines. Additional articles were included from perusing existing local publications and reference lists. An additional search was also completed for neurologists with hyphenated names using both maiden and married names. Deceased Filipino neurologists were sourced from the PNA website and directory and also included in the search. We excluded studies that were authored by PNA neurologists but did not have an affiliation in the Philippines at the time of publication, conference proceedings, letters to the editor or correspondence, editorials or commentaries, and guidelines.

Duplicates were identified and excluded through juxtaposition of study titles, authors, and year of publication. Eligibility was assessed by two reviewers (ADOA and RDGJ) after duplicates were removed. Disagreements were resolved by discussion and consensus. Studies fulfilling the eligibility were included in the quantitative analysis.

Extracted articles were encoded and archived into a Microsoft Excel file including: (a) year of publication, (b) authors, (c) affiliations, (d) journal of publication, (e) 2021 journal impact factor, (f) study design, (g) main topic of research. Citation count of articles on Scholar, CrossRef, Web of Science, Dimensions, and Scopus was manually collected, and the citation count was determined by the highest value. Journal impact factor was determined from SCImago Journal and Country Rank [20].

A list of Filipino adult and child neurologists practicing in the Philippines was compiled using the PNA website. The list was further organized by region of practice and field of practice (adult, child, or both). Region-specific socioeconomic parameters (gross domestic product (GDP), population, health expenditure, number of health establishments, %GDP allocated for research and development (R&D), and number of R&D personnel) were sourced from official government reports under the Philippine Statistics Authority [21,22] and the Compendium of Science and Technology [23]. Health expenditure is determined from health financing revenues using the Philippine National Health Accounts framework. Research expenditure is based on the %GDP allocated for R&D.

Data syntheses were conducted in Microsoft Excel and R version 4.0.3 [24] using descriptive statistics such as frequencies and percentages. Spearman rank-order correlation was used to measure the association between regional socioeconomic indicators and bibliometric indices. *p*-value was set at 0.05.

## 3. Results

### 3.1. General Information

A total of 1232 articles were retrieved from an electronic database search and an additional 78 articles were identified from a manual review of locally published journals. We removed 38 duplicate records prior to screening. Screening of 1272 article titles and abstracts yielded 314 studies for exclusion. These included conference proceedings and guidelines. The full text could not be retrieved for 29 articles. The 929 remaining full-text articles were assessed for eligibility. A total of 746 studies were included for quantitative analysis. All retrieved studies were published in the English language (Figure 1).

The 746 studies were published in 245 journals and books from 1966 to 2021, amassing a total of 12,409 citations. Each cited article had an average of 17 and a median of 8 citations (range 0–511). Of the 746 studies, 302 (40.5%) were uncited. The most cited article was a cohort study about juvenile myoclonic epilepsy published in *Neurology* in 1984, garnering 511 citations [25]. The second most cited article had 408 citations and was a pre-clinical study on botulinum toxin published in *Muscle and Nerve* in 1996 [26]. The year 2021 saw the greatest number of publications, with 116 new publications indexed that year.

### 3.2. Proportion of Publications and Citations Per Region

The majority of the research output and citations came from the National Capital Region (NCR). NCR publications comprised 88.7% (*n* = 662) of Philippine publications and 83.6% (*n* = 10,377) of total citations (Figure 2). CALABARZON (Region IVA) had the second highest number of publications by a wide margin (*n* = 79, 10.6%), followed by Western Visayas (Region VI) (*n* = 43, 5.8%). Davao Region (Region XI) had the second highest number of citations (*n* = 1089, 8.8%) closely followed by Western Visayas (Region VI) (*n* = 1021, 8.2%). NCR had the most inter-regional collaborations (*n* = 15), followed by Region IVA (*n* = 11) and CAR (*n* = 10). Regions XII, IVB, VIII, and XIII had the least.

### 3.3. Publications by Adult and Child Neurologists and Institutions

Among the 513 practicing neurologists in the country, 435 are adult neurologists and 84 are child neurologists. Five of them are board-certified in both adult and child neurology. An additional 14 neurologists have passed on, 2 of which were child neurologists. More than half of adult neurologists (*n* = 221, 50.8%) and child neurologists (*n* = 53, 63.1%) have at least one publication. Adult neurologists have published 460 papers internationally and 210 locally. For pediatric neurologists, the number is 38 internationally and 53 locally (Figure 3).

The institutions with the most publications are the *Philippine General Hospital, University of the Philippines Manila* (*n* = 337, 45.2%), *University of Santo Tomas* (*n* = 199, 26.7%), *St. Luke’s Medical Center* (*n* = 141, 18.9%), *Philippine Children’s Medical Center* (*n* = 64, 8.6%), *Makati Medical Center* (*n* = 36, 4.8%), and *Jose Reyes Memorial Medical Center* (*n* = 35, 4.7%). These institutions also had the most citations: *University of Santo Tomas* (*n* = 5577, 44.9%), followed by the *Philippine General Hospital*, *University of the Philippines Manila* (*n* = 4351, 35.1%), and *St. Luke’s Medical Center* (*n* = 1751, 14.1%). These top-cited institutions are all located in the NCR.

The majority of studies were completed between at least two institutions. Of the 746 studies, 314 (42.1%) were single-institution studies, and 521 (69.8%) were initiated by a local institution. With international collaborators, the greatest number of articles were published with the *National University of Singapore (n* = 49, 6.6%*), University of Luebeck, Germany* (*n* = 29, 3.9%)*, National Neuroscience Institute of Singapore* (*n* = 27, 3.6%), *Seoul National University* (*n* = 25, 3.4%), *Mahidol University of Thailand* (*n* = 23, 3.1%)*,* and *Massachusetts General Hospital* (*n* = 23, 3.1%). There were 88 affiliations from the United States, 43 from Germany, and 38 from Japan.

### 3.4. Article Characteristics

The most common topics of research were closely tied between stroke and/or cerebrovascular disease (*n* = 183, 24.5%) and movement disorders (*n* = 165, 22.1%). Neurologic infections (*n* = 67, 9.0%) were the next most common followed by neuro-immunology and autoimmune diseases (*n* = 49 each, 6.6%), dementia (*n* = 47, 6.3%), and epilepsy (*n* = 42, 5.6%). There were 23 (3.1%) studies about nerve disorders and 21 (2.8%) studies about neurodevelopmental or genetic disorders and neuromuscular disorders, respectively. The remaining topics were neuro-oncology (*n* = 18, 2.4%), neuropsychiatry (*n* = 17, 2.3%), neurotoxicology (*n* = 16, 2.1%), headache and pain (*n* = 13, 1.7%), and congenital disorders (*n* = 11, 1.5%) (Figure 4).

Case reports and case series (*n* = 183, 24.5%) were the most common, followed by cohort studies (prospective and retrospective) (*n* = 158, 21.2%) and cross-sectional studies (*n* = 126, 16.9%). There were 59 reviews (7.9%), 51 systematic reviews and/or meta-analyses (6.8%), 48 clinical trials (6.4%), 33 case–control (4.4%), 32 animal or laboratory studies (4.3%), and 27 book chapters (3.6%) (Figure 4).

### 3.5. Journals, Impact Factor, and Citations

A total of 235 journals were identified, 24 of which were local journals. Most articles (*n* = 491, 65.8%) were published in international journals, accounting for the majority of citations (*n* = 12,240, 98.6%). More than 1 out of every 10 articles was published locally in the *Philippine Journal of Neurology* (*n* = 86, 11.5%). Nearly one-fourth of indexed articles were published in local journals including *Acta Medica Philippina* (*n* = 64, 8.6%), the *Philippine Journal of Internal Medicine* (*n* = 28, 3.8%)*,* and the *University of Santo Tomas Journal of Medicine* (*n* = 14, 1.9%). The top publishing international journals were *Parkinsonism and Related Disorders* (*n* = 19, 2.5%)*, Neurology Asia* (*n* = 17, 2.3%)*,* and *Stroke* (*n* = 12, 1.6%).

As of 2021, nearly half of the journals had an impact factor (IF) of more than 2 (*n* = 114, 48.5%). The highest impact journals where Filipino neurologists published were *Cell* (IF 45), *The Lancet* (IF 22.23), *Lancet Neurology* (IF 15.9), *Nature Communications* (IF 15.41), and *Science* (IF 15.19). Only 34 articles were published in the top 10% of journals by IF, which was less than 5% of the total number of indexed articles.

The most cited journals were *Stroke* (IF 5.89, 1099 citations), which published 12 articles, *Neurology* (880 citations, 8 articles), *Annals of Neurology* (670 citations, 5 articles), *Muscle & Nerve* (514 citations, 4 articles), and the *Journal of Neural Transmission* (448 citations, 11 articles).

### 3.6. Socioeconomic Factors, Health, and Research Expenditure

As of 2020, the most populated regions are Region IVA (CALABARZON), NCR, and Region III (Central Luzon). Together these comprise 38.6% of the total Philippine population. The least populated regions are Region IVB (MIMAROPA), Region XIII (CARAGA), and the Cordillera Administrative Region (CAR). GDP was highest in NCR, Region IVA, and Region III and lowest in CAR, Region XIII, and the Autonomous Region in Muslim Mindanao (ARMM).

According to the Philippine Statistics Authority, the number of healthcare establishments was greatest in NCR, Region IVA, and Region III; and the least in CAR, Region XIII, and ARMM. Health expenditure in 2020 was greatest in NCR, Region III, and Region VI (Western Visayas). It was lowest in Region XIII, ARMM, and Region IVB. As of 2022, there are 513 adult and child neurologists. Most neurologists are based in NCR (*n* = 236, 46.0%), Region IVA (*n* = 71, 13.8%), and Region III (*n* = 41, 8.0%). Region IX (Zamboanga), Region XIII, and ARMM only have four, three, and two neurologists, respectively, which altogether is less than 2% of the total number of neurologists in the country. However, in terms of proportion of neurologists with at least one publication, Region IX (75.0%), XIII (66.7%), and CAR (66.7%) had more published neurologists compared to NCR (65.6%).

Research expenditure in 2018 was determined to be highest in ARMM (0.7% of regional GDP), Region III (0.35%), and Region IVA (0.2%), and lowest in Region I (Ilocos Region) (0.05%), Region XII (SOCCSKSARGEN) (0.03%), and Region IVB (0.03%). In terms of overall personnel for research and development, NCR, Region IVA, Region III have the most and Region XIII, XII, ARMM have the least (Table 1).

### 3.7. Correlation among Bibliometric Indices and Socioeconomic, Health, and Research Parameters

The correlation of bibliometrics with socioeconomic, health, and research parameters are visually summarized in Figure 5. The number of total publications, local publications, international publications, the number of total citations, and the number of published neurologists were positively associated with the population (*p* = 0.001 to *p* = 0.020), regional gross domestic product (*p* < 0.001 to *p* = 0.002), number of healthcare establishments (*p* < 0.001 to *p* = 0.005), healthcare expenditure (*p* = 0.007 to *p* = 0.039), number of personnel in research and development (*p* < 0.001 to *p* = 0.002), and the number of neurologists (*p* < 0.001 to *p* = 0.003). The ratio of published to unpublished neurologists was negatively associated with the population (*p* = 0.169), regional GDP (*p* = 0.322), number of healthcare establishments (*p* = 0.323), health expenditure (*p* = 0.155), and number of personnel in research and development (*p* = 0.351); however, these were not statistically significant. %GDP for research and development did not show a significant correlation with any of the variables (*p* = 0.313 to *p* = 0.799) (See Appendix A).

## 4. Discussion

Research activity in Southeast Asia has been shown to be predominated by countries with higher income such as Singapore, Thailand, and Malaysia [6]. While the Philippines is an LMIC, our findings showed steady growth in research output, amounting to 746 articles in the span of 45 years. Research output is measured not only by the number of publications but also in terms of impact and dissemination, which is quantified by the number of citations [27]. The Philippines may be publishing more, as shown by the marked increase in publications in 2021, likely owing to the COVID-19 pandemic. Restructuring of the healthcare workforce and mandatory quarantine during the pandemic reduced clinical encounters and obligations [28], which may have allowed neurologists more time to conduct research, particularly those that did not require direct patient interaction such as systematic reviews and/or meta-analyses and studies involving retrospective chart review. There was likely more time to complete ongoing research manuscripts. Most studies published in 2021 were case reports and case series (*n* = 38), systematic reviews and reviews (*n* = 20), cohort studies (*n* = 19), and cross-sectional studies (*n* = 13).

Despite the increase in the number of publications, 4 out of every 10 studies remain uncited. These studies were mostly local publications and recently published studies that would reasonably have fewer or no citations yet. Additionally, indices for local publications and the studies retrieved through printed text do not have embedded counters for citations, which may underestimate the total reach of these studies.

At least one in two Filipino neurologists has been published. There are disproportionately more adult neurologists in the Philippines than there are pediatric neurologists. This may be in part due to the number of years required to complete training and the number of training slots available. Any licensed physician may apply for Adult Neurology residency, but to be a pediatric neurologist, one must complete Pediatrics or Adult Neurology residency in order to be eligible for fellowship training. There are also more institutions offering Adult Neurology residency training (*n* = 11) than there are institutions offering Pediatric Neurology fellowship training (*n* = 3). In general, more years are spent in residency (3 to 4 years) compared to fellowship (3 years) with more training slots offered for residency compared to fellowship. This creates a larger pool of trainees in total per year, which would evidently produce more graduates and also afford more opportunities for senior neurologists to collaborate and co-author research publications. Despite this, the percentage of published child neurologists was greater than that of adult neurologists.

Most of the publications and citations can be traced to research output from NCR. This is likely due to the fact most of the training institutions and neurologists are based here [29]. Correlation analysis (Figure 5) showed a very strong positive association between the number of neurologists (*R* = 0.98, *p* < 0.001) and the number of affiliated institutions (*R* = 0.99, *p* < 0.001) with the total number of publications regardless, if local or international (See also Appendix A). The study has also found that the local institutions with the most publications are also located in these regions and also happen to be training institutions. Training institutions tend to have a greater drive to publish, given that publication is a requirement for promotion and/or graduation. The research output of Region IVA likewise may have benefited from its proximity to NCR where it is common for neurologists in this area to affiliate with institutions belonging to both NCR and Region IVA. A significant bulk of research output from Western Visayas may be attributed to Panay Island, the geographic origin of X-linked dystonia-parkinsonism (XDP), a movement disorder endemically unique to this region [30].

Similar to the trend observed in the bibliometrics for neurologic diseases in Southeast Asia, the majority of studies were case reports, case series, cohort, and cross-sectional studies. Altogether, these constitute more than half of the publications. The frequent utilization of these types of studies may be due to fewer resources, manpower, and organization required in comparison to larger, experimental studies [31]. There may also be a novelty in rare disease presentations in an underrepresented region and population for international journals.

It is interesting to note that there were almost just as many published studies on cerebrovascular disease and movement disorders. In Southeast Asia, the Philippines ranks 3rd for stroke incidence [32]. Stroke is also the second leading cause of mortality in the country [33]. In a bibliometric analysis of stroke research output, the Philippines contributed only 5.9% to stroke research productivity in Southeast Asia, although the disease burden was not deemed to be correlated with the number of publications or citations. Regardless, there exists a wealth of study population due to its high incidence in the country, which may facilitate research endeavors on this topic.

In the case of movement disorders, the Philippines contributed to 45.9% (*n* = 68) of research outputs in Southeast Asia about dystonia, ranking first in the region. This is a stark contrast to the country’s research contributions to motor neuron disease, headache, multiple sclerosis, neuro-oncology, and dementia, ranging from 1–6% of total publications [7,9,12,13]. This may be attributed to growing research on XDP, which was first described in the literature in 1975 [30]. This has since sparked international interest and collaboration leading to studies on symptomatology, natural history, and treatment for this endemic, hereditary neurodegenerative disease.

Institutional and international collaboration has been propagated in recent years [34]. One-way international linkages are formed when graduates of Philippine neurology seek further training abroad. This was seen in the number of studies co-authored by a Filipino neurologist yet excluded from analysis as they were not affiliated with a local institution at the time of publication.

Single-authored papers are a rare minority (*n* = 25, 3.4%), yet almost half of the studies identified had authors belonging to only one institution (*n* = 314, 42.1%). Leading collaborators based on the number of co-written articles were from Germany, Singapore, South Korea, Thailand, and the United States. Although foreign-led studies only comprised 29.8% of total publications, these studies were co-authored by institutions in countries belonging to a higher income stratum, which may guarantee greater funding and better success in international publication [35]. Additionally, international publications accounted for the majority of citations, as these journals would have a wider reach in comparison to local journals.

In contrast, it was observed that local publications had comparatively lower citation counts. This may in part be due to the lack of online documentation and citation metrics embedded in these local journals, especially in their print form. Obtaining readership and research impact has been identified as a challenge, but this has been precluded in this age of digital media by setting up websites and allowing open access for articles under these journals [36]. Additionally, efforts have been made to have local journals indexed in international databases, allowing their contents to be readily searched. Among the most published local journals, *Acta Medica Philippina* is indexed internationally in Scopus, Google Scholar, ASEAN Citation Index (ACI), and Western Pacific Region Index Medicus [37]. The *Philippine Journal of Internal Medicine* was previously indexed in Scopus from 1973–1995 and 1997 [20,36].

The scarcity of indexed local journals may in part be due to the challenge in meeting international standards, which is grossly dependent on the number and quality of submissions. However, despite local policy changes offering incentives for improvement of local journal quality since 2009 [38] the appeal to publish internationally prevails, as evidenced in the more than three-fold increase (338.4%) in international publications in the last decade compared to 24.7% growth in the number of local studies published between 2012 to 2021 compared to 2002 to 2011 (Figure 2). The incentive for local journals may be overshadowed by other policies such as the International Publication Award offered by institutions such as the University of the Philippines System, University of Santo Tomas, and the Department of Science and Technology, which provides a cash incentive proportional to the IF of the journal of publication [39]. Interestingly, the majority of publications by adult neurologists were published in international journals while publications by child neurologists were predominantly in local journals. Investigating perceptions and attitudes towards research publication and factors determining the journal for publication may be a direction for further study. It may be of note that all three pediatric neurology training institutions have their own institutional local publication, which may pose as more accessible and affordable options for research dissemination.

Publication in high-impact journals is uncommon. Despite the majority of studies being published in international journals, only 5% of studies were accepted in journals with the top 10% highest impact factors. A survey evaluating determinants of publication in high-impact journals found a higher likelihood for doctorate holders compared to medical doctors; older authors; more time and resources devoted to research; and English as a first language [40]. The Philippines uses English primarily in formal education and in the practice of medicine; however, the majority of the Filipino neurologists identified in this study did not have Ph.D. degrees. The overwhelming burden of neurological disease in the country may also take away dedicated time for research toward patient care. In addition, as a low–middle income country, there is more to be desired when it comes to budget allocation for research, although R&D expenditure has increased from 0.16% in 2015 to 0.32% in 2018 [23]. Correlational analysis in this study however does not establish a statistically significant association with allocation for R&D with any of the bibliometric indices. This is contradictory to findings from previous literature which found a positive association between higher R&D spending and total publications and citations [41]. This is possibly explained by outlier data from ARMM, which has a 0.7% allocation for R&D yet is one of the regions with the least number of neurologists and publications.

For this study, in the absence of regionally distributed data on incidence, the burden of disease was grossly approximated by the number of neurologists and healthcare establishments (healthcare access and delivery) and the amount of healthcare expenditure (healthcare utilization). Analysis showed a positive correlation between the aforementioned factors and research productivity. Based on this, areas with a greater burden of disease will have a larger patient population, lending to more research opportunities for descriptive studies. These areas with more doctors and hospitals would also more likely belong to an urban area where general income would be higher, and income has already been shown to be correlated with research output.

In summary, there were more publications and citations in more populated areas with higher income, better access to healthcare establishments, and more neurologists. At present, the bulk of training institutions and neurologists are concentrated in NCR [18]. This study has identified a stark disparity between the NCR and other regions in terms of research productivity and resources for health and research. Small steps have been undertaken to address this through the establishment of training programs in other regions, such as in CAR and Region VII (Central Visayas). Another direction for improvement would be promoting multi-institutional studies locally. A recently completed and timely example would be the Philippine CORONA study which was able to gather a total of 10,881 patients from 37 institutions nationwide, including both training and non-training institutions representing both the private and government sectors [42]. Similarly, the development of a centralized registry for different neurological diseases could prove to be an opportunity for collaboration between the different contributing training institutions [43].

This study has quantified 45 years of research output of Filipino adult and child neurologists. This is the second known aggregation of studies specific to Philippine neurosciences, and it has surpassed its predecessor in terms of scope and content. The year of inclusion is twice as long and contains more than seven times the original number of published articles. This study provides an updated and comprehensive glimpse of the trends in research in Philippine neurology.

This study is limited in that only published studies were accounted. There are numerous studies completed each year in the country; however, only a percentage of which are published. Further research on the characteristics of unpublished completed studies may be pursued. In addition, neurologists were indexed according to their assigned region on the PNA website without redundancy between regions. In reality, there are neurologists based in other regions with visiting practices in NCR or nearby regions and vice-versa. The total number of neurologists also remains an approximation as a number of them are currently in training abroad and may not be contributing to the local health workforce. Identification of publications by female neurologists was doubly difficult as a thorough search using their maiden name, married name, and the hyphenated name was required. Likewise, conscientious screening of studies by neurologists with common surnames needed verification based on affiliation.

## 5. Conclusions

Research productivity in Philippine neurology has shown steady growth in recent years. The research landscape is dominated by studies authored by neurologists belonging to institutions in the NCR, which has the greatest number of neurologists, training institutions, and GDP. Greater research productivity was associated with population, GDP, health expenditure, number of healthcare establishments, neurologists, and research personnel. There is a need to address the disparity seen in more distant regions in order to bridge gaps in healthcare, health human resources, and health information through research.

## Figures and Tables

**Figure 1 ijerph-19-15630-f001:**
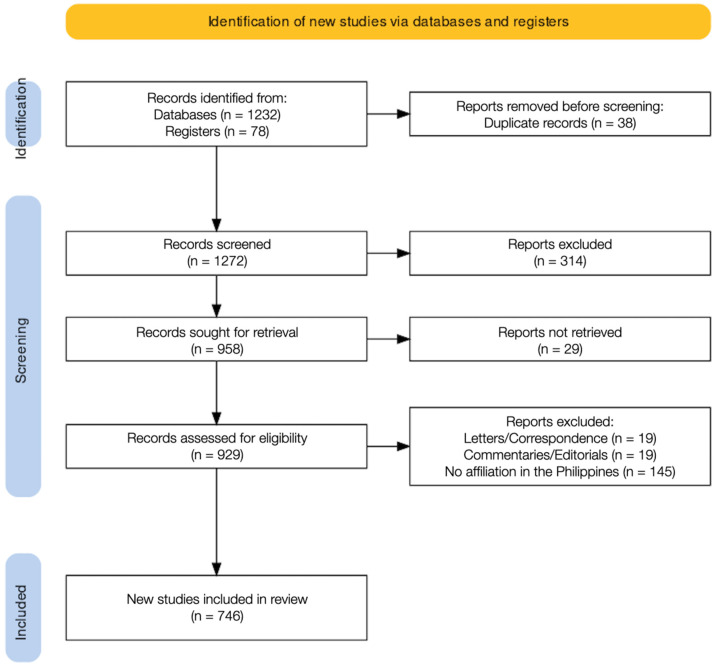
PRISMA flow diagram of study selection.

**Figure 2 ijerph-19-15630-f002:**
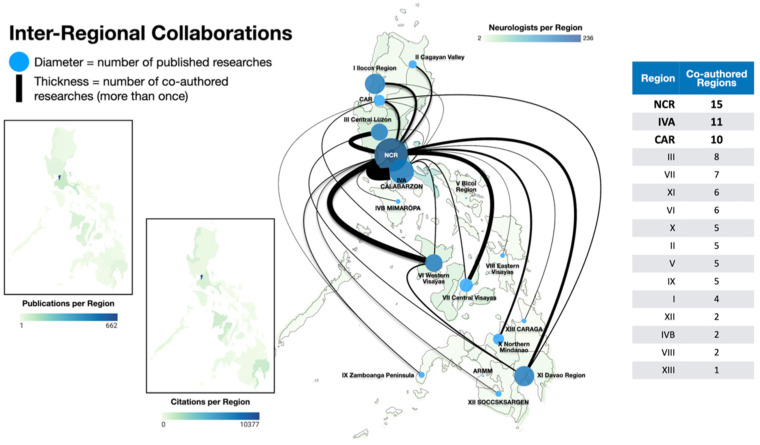
Philippine national map with inter-regional collaboration networks and choropleth of bibliometric indices and number of neurologists composited.

**Figure 3 ijerph-19-15630-f003:**
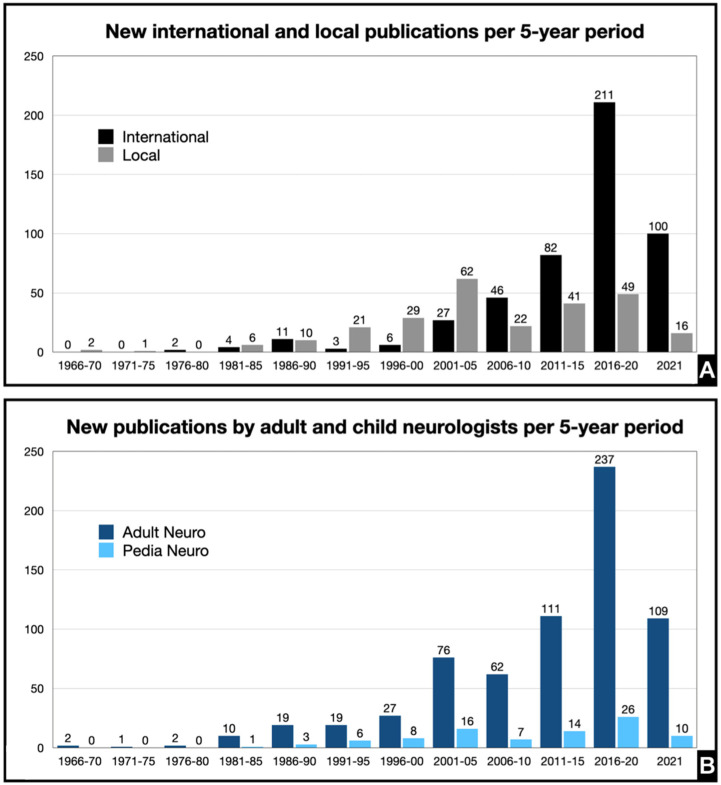
Trends in research publications of Filipino neurologists per 5-year period by journal of publication (**A**) and by field of practice (**B**).

**Figure 4 ijerph-19-15630-f004:**
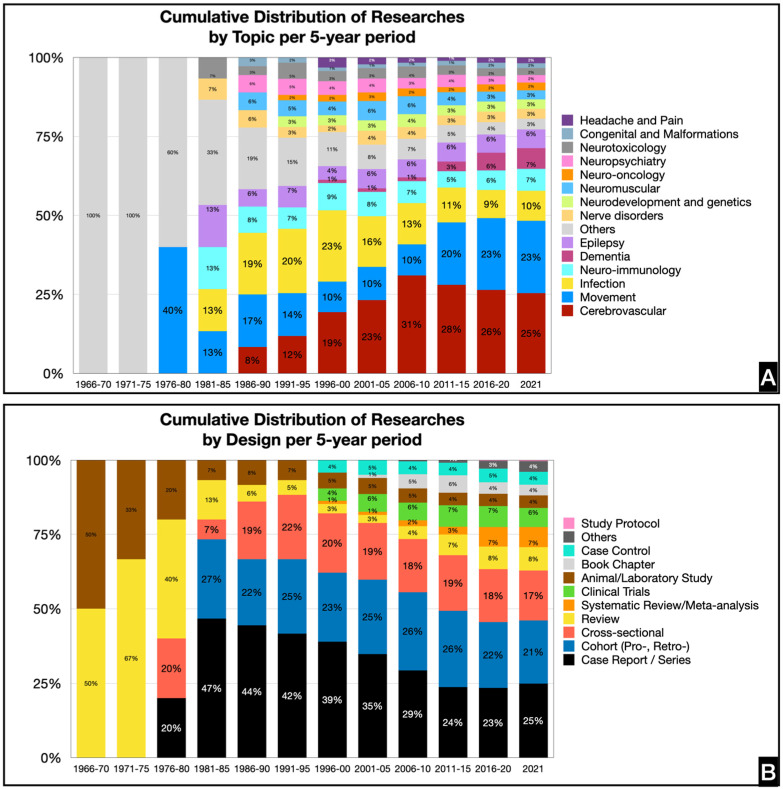
Trends in research publications of Filipino neurologists per 5-year period by research topic (**A**) and by research design (**B**).

**Figure 5 ijerph-19-15630-f005:**
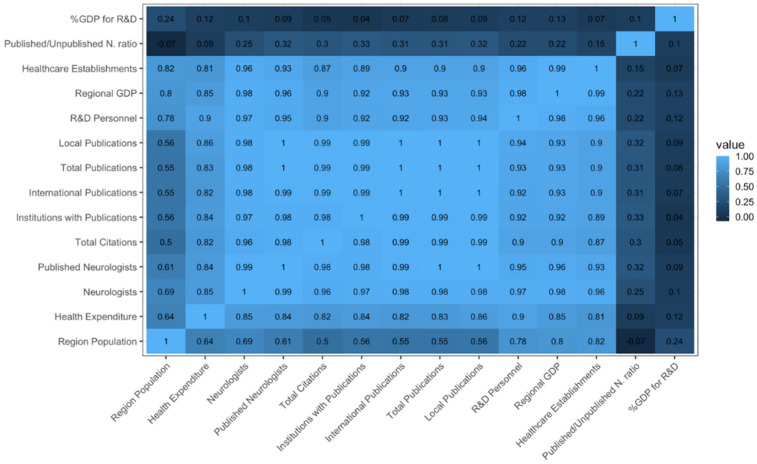
Correlation heatmap of bibliometrics, socioeconomic, healthcare, and research indices. Values closer to 1 have greater correlation.

**Table 1 ijerph-19-15630-t001:** Bibliometric indices, socioeconomic factors, healthcare and research indices of the Philippines and individual regions.

	Bibliometric Indices												Health Indices				Socioeconomic Indices		Research Indices	
	Total Publications		International Articles		Local Articles		Total Citations		Institutions with Publications		Neurologists with Publications		Total Living Neurologists		Health Expenditure ^a^	Health Establishment ^b^	Population ^c^	GDP ^d^	%GDP for R&D ^e^	R&D Personnel ^e^
Philippines	746	%	491	%	255	%	12,409	%	80	%	274	52.1	513	%	209.59	1096	109.0	17,537.8	0.32	75,036
NCR	662	88.7	437	89.0	225	88.2	10,377	84.6	45	56.3	160	65.6	236	46.0	155.30	322	13.5	5599.9	0.19	22,285
CAR	11	1.5	10	2.0	1	0.4	182	1.4	1	1.3	6	66.7	8	1.6	19.12	14	1.8	288.9	0.07	2057
I—Ilocos Region	25	3.4	20	4.1	5	2.0	756	5.7	0	0.0	6	24.0	25	4.9	39.57	46	5.3	581.9	0.05	2995
II—Cagayan Valley	8	1.1	4	0.8	4	1.6	175	1.3	0	0.0	7	50.0	14	2.7	23.74	29	3.7	371.1	0.11	2976
III—Central Luzon	28	3.8	10	2.0	18	7.1	226	1.7	2	2.5	17	40.5	41	8.0	98.37	100	12.4	1881.3	0.35	8698
IVA—CALABARZON	79	10.6	58	11.8	21	8.2	734	5.6	6	7.5	30	42.3	71	13.8	20.81	177	16.2	2534.4	0.2	8724
IVB—MIMAROPA	3	0.4	3	0.6	0	0.0	279	2.1	1	1.3	2	40.0	5	1.0	7.70	15	3.2	357.8	0.03	1188
V—Bicol Region	1	0.1	1	0.2	0	0.0	5	0.04	0	0.0	1	10.0	9	1.8	38.45	37	6.1	517.8	0.12	2264
VI—Western Visayas	43	5.8	24	4.9	19	7.5	1021	7.8	8	10.0	12	38.7	31	6.0	58.74	46	8.0	825.4	0.11	5694
VII—Central Visayas	33	4.4	20	4.1	13	5.1	267	2.0	6	7.5	13	48.1	26	5.1	50.72	73	8.1	1134.9	0.07	4979
VIII—Eastern Visayas	3	0.4	2	0.4	1	0.4	10	0.1	0	0.0	2	40.0	5	1.0	24.93	19	4.5	434.8	0.08	2893
IX—Zamboanga	6	0.8	4	0.8	2	0.8	137	1.0	2	2.5	3	75.0	4	0.8	21.70	23	3.9	376.3	0.1	1462
X—Northern Mindanao	11	1.5	5	1.0	6	2.4	47	0.4	1	1.3	5	31.3	15	2.9	30.10	62	5.0	821.4	0.06	3104
XI—Davao Region	25	3.4	21	4.3	4	1.6	1089	8.3	4	5.0	5	45.5	11	2.1	35.06	61	5.2	833.2	0.06	3159
XII—SOCCSKSARGEN	4	0.5	3	0.6	1	0.4	54	0.4	3	3.8	2	28.6	7	1.4	35.06	58	4.9	449.2	0.03	896
XIII—CARAGA	3	0.4	1	0.2	2	0.8	12	0.1	1	1.3	2	66.7	3	0.6	15.31	11	2.8	285.0	0.08	1061
ARMM	1	0.1	1	0.2	0	0.0	0	0.0	0	0.0	1	50.0	2	0.4	14.59	3	4.4	244.5	0.7	601

^a^ In thousands Philippine pesos as of 2020; ^b^ number of health establishments, as of 2017; ^c^ population in millions, as of 2020; ^d^ in millions Philippine pesos, at constant 2018 prices, as of 2020; ^e^ as of 2018; GDP—gross domestic product; NCR—National Capital Region; CAR—Cordillera Administrative Region; ARMM—Autonomous Region in Muslim Mindanao.

## Data Availability

The raw data supporting the conclusions of this article will be made available by the authors upon reasonable request.

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
