# Peer review of "Research Productivity among Filipino Neurologists Associated with Socioeconomic, Healthcare, and Disease Burden Factors: A Bibliometric Analysis"

_ijerph, 2022, doi:10.3390/ijerph192315630_

Round 1

Reviewer 1 Report

Thank you for the opportunity of reviewing this paper.

The topic is interesting – as a bibliometric analysis, it reflects the lack of equity for access to contribute to academic research in certain regions of the Philippines.

I have several comments that can help improve the manuscript.

Of note, I could not find Figure 5; please make sure to provide it when submitting your revision.

Title:

The title is well-written and addresses the topic of the paper.

Abstract:

-       Line 19: Please rephrase this sentence for further clarity: “The total publications (n = 662, 88.7%) and citations (n = 10,377, 83.6%) were highest in the National Capital Region.”

-       Lines 21-23: positively correlated with the number of neurologists, research personnel? Please be more clear with the variables analyzed.

Introduction:

-       Please try to adjust the introduction, so it flows better before presenting the paper's goal; you can consider starting with the geographical - sociological background of the Philippines, then move to the status of neurology training there, and finally raise concern about research productivity and how it is measured.

-       Lines 32-33: please rephrase this sentence for further clarity.

-       Lines 38-39: please rephrase this sentence for further clarity.

-       Lines 43-44: do you mean “compiled neuroscience research studies”?

-       Lines 63-71:  I would move this to the Discussion.

Methods:

-       Lines 84-85: Why did you exclude studies that were not in English language? This could be a bias. It would be interesting to know about the scientific literature in other Filipino/Tagalog.

-       Lines 90-93: What kind of disagreements did you encounter?

-       Lines 102-104: Please specify the categories for socioeconomic parameters in the Methods.

-       Please mention here the exclusion criteria. You said that you were including any study design but then you excluded letters, editorials, etc. Please clarify.

-       Please define the terms “research expenditure” and “health expenditure” in the Methods.

Results:

-       Lines 112-118: were all the articles found electronically?

-       Figure 1: please explain the abbreviations in the figure’s footnote (what’s PH?).

-       Lines 124-126: Can you mention what was the most cited article in the fields of adult neurology and child neurology separately?

-       Lines 124-126: Who was the most cited author?

-       It would be interesting to add the results by gender if you have these data, although I am aware this is limitation that you address in the Discussion.

-       Section 3.7: Please add p values to every statement in this section.

-        Where is figure 5?

Discussion:

-       Lines 21-23: How would you explain that there are more publications because of COVID?

-       Line 43: The figure 4 cited here does not reflect a correlation analysis.

-       Lines 67 and 71: there are random numbers here, are they perhaps references?

-       Lines 72-74: expand more on the marked trend in publications about movement disorders compared with other countries in South Asia. Can you compare your numbers with those of other countries in the world as well? Do you think the Philippines has a special potential in this research topic?

-       In general, the Discussion could be substantially shortened to highlight the most relevant results, without repeating information.

Author Response

REVIEWER #1

Comments/Recommendations

Response

Thank you for the opportunity of reviewing this paper.

The topic is interesting – as a bibliometric analysis, it reflects the lack of equity for access to contribute to academic research in certain regions of the Philippines.

I have several comments that can help improve the manuscript.

Of note, I could not find Figure 5; please make sure to provide it when submitting your revision.

Thank you for bringing this to attention.

Figure 5 has been appended to the manuscript.

Please see Page 10 Line 4.

TITLE:

The title is well-written and addresses the topic of the paper.

Thank you very much.

ABSTRACT:

Line 19: Please rephrase this sentence for further clarity: “The total publications (n = 662, 88.7%) and citations (n = 10,377, 83.6%) were highest in the National Capital Region.”

Thank you very much for your inputs. Phrasing has been revised and hopefully provides more clarification:

“The National Capital Region (NCR) had the most publications (n = 662, 88.7%) and citations (n = 10,377, 83.6%).”

Please see page 1, lines 19-20.

ABSTRACT:

Lines 21-23: positively correlated with the number of neurologists, research personnel?

Thank you very much. Research productivity had positive correlation with the number of healthcare establishments, number of neurologists, and number of research personnel. The Oxford comma was used in this sentence specifying healthcare establishments, neurologists, and research personnel under “number”.

Please see page 1, lines 21-22.

INTRODUCTION:

Please try to adjust the introduction, so it flows better before presenting the paper's goal; you can consider starting with the geographical - sociological background of the Philippines, then move to the status of neurology training there, and finally raise concern about research productivity and how it is measured.

Thank you very much for your suggestions. Paragraph order has been revised to facilitate better flow of ideas.

Please see page 1, lines 31-36.

INTRODUCTION:

Lines 32-33: please rephrase this sentence for further clarity.

Thank you for your inputs. This has been revised in the Introduction as:

“A comparison of DALYs for neurological diseases between 1990 and 2016 showed an 87% increase, which highlights a growing demand for neurologists in the country.”

Please see page 1, lines 37-40.

INTRODUCTION:

Lines 38-39: please rephrase this sentence for further clarity.

Thank you for noticing this. We have rephrased the sentence for further clarity:

“The official PNA website lists approximately five hundred adult and pediatric neurologists in the country.”

Please see page 2, lines 45-46.

INTRODUCTION:

Lines 43-44: do you mean “compiled neuroscience research studies”?

Thank you very much. That is indeed what the sentence is referring to and has been rephrased as per suggestion.

“In 2007, BRAIN Inc. compiled neuroscience research studies from 1985 to 2006 in the country in a compendium…”

Please see page 2, lines 49-51.

INTRODUCTION:

Lines 63-71:  I would move this to the Discussion.

Thank you very much for this suggestion. Lines 63-71 in the original manuscript were findings from review of related literature, which we wanted to mention in the Introduction in order to build points towards significance of the study.

Please see page 2, lines 62-68.

METHODS:

Lines 84-85: Why did you exclude studies that were not in English language? This could be a bias. It would be interesting to know about the scientific literature in other Filipino/Tagalog.

Thank you very much for bringing this to attention. Formal communication in the Philippines is conducted primarily in English thus all research publications are written in English. The lack of Filipino language scientific literature indeed is an interesting point of note. A sentence referring to this has been added to the Results and inclusion criteria has been revised under Materials and Methods.

“All retrieved studies were published in the English language.”

Please see page 3, line 122.

METHODS:

Lines 90-93: What kind of disagreements did you encounter?

Thank you for your interest! There was some contention with regards to large multicenter studies wherein searching for an author’s name will bring up results for a large study which they contributed to as a site investigator but were not among the first authors. We opted to not count these studies under these authors because as per JCEM authorship guidelines, they were contributors to data collection but did not significantly contribute towards authorship of the manuscript.

METHODS:

Lines 102-104: Please specify the categories for socioeconomic parameters in the Methods.

Thank you very much for your suggestion.

We have rephrased this line to include the categories of socioeconomic parameters:

“Region-specific socioeconomic parameters (gross domestic product (GDP), population, health expenditure, number of health establishments, %GDP allocated for research and development (R&D), and number of R&D personnel)

were sourced from official government reports under the Philippine Statistics Authority [21,22] and the Compendium of Science and Technology [23].”

Please see page 3, lines 102-104.

METHODS:

Please mention here the exclusion criteria. You said that you were including any study design but then you excluded letters, editorials, etc. Please clarify.

Thank you very much for pointing this out. An additional sentence specifying exclusion has been included under Materials and Methods.

“We excluded studies that were authored by PNA neurologists but did not have an affiliation in the Philippines at the time of publication, conference proceedings, letters to the Editor or correspondence, editorials or commentaries, and guidelines.”

Please see page 2, lines 86-89.

METHODS:

Please define the terms “research expenditure” and “health expenditure” in the Methods.

Thank you very much. We have added an additional statement under Materials and Methods with the definition:

“Health expenditure is determined from health financing revenues using the Philippine National Health Accounts framework. Research expenditure is based on the %GDP allocated for R&D.”

Please see page 3, lines 106-108.

RESULTS:

Lines 112-118: were all the articles found electronically?

Thank you for your inputs. Majority were found digitally (1232), and 78 were retrieved from print.

RESULTS:

Figure 1: please explain the abbreviations in the figure’s footnote (what’s PH?).

Thank you for pointing this out. We have fixed Figure 1, spelling out PH. We meant to refer that the author did not have an affiliation in the Philippines.

RESULTS:

Lines 124-126: Can you mention what was the most cited article in the fields of adult neurology and child neurology separately?

Thank you very much for this suggestion!

We have added additional information for this section. We also noticed a typographical error regarding the most cited article. It should have 511 citations instead of only 51.

“The most cited article was a cohort study about juvenile myoclonic epilepsy published in the Neurology in 1984, garnering 511 citations [25]. The second most cited article had 408 citations and was a pre-clinical study on botulinum toxin published in Muscle and Nerve in 1996 [26].”

Please see page 4, lines 128-131.

RESULTS:

Lines 124-126: Who was the most cited author?

Thank you for your interest. Our focus in this study was on regional rather than individual research productivity, thus we opted to withhold results for the most cited authors and instead approximated this by reporting institutions with the most citations and publications.

RESULTS:

It would be interesting to add the results by gender if you have these data, although I am aware this is limitation that you address in the Discussion.

Thank you very much for your understanding. This may be a direction for further study. Although biological sex may be assumed based on the names of the neurologists on the website, it is challenging to determine gender based on this alone.

RESULTS:

Section 3.7: Please add p values to every statement in this section.

Thank you very much for pointing this out. We have revised Section 3.7 to include the p values of correlation analysis. We have also attached a Supplementary File containing the correlational analysis results with p values.

“The correlation of bibliometrics with socioeconomic, health, and research parameters are visually summarized in Figure 5. The number of total publications, local publications, and international publications, the number of total citations, and the number of published neurologists were positively associated with population (p = 0.001 to p = 0.020), regional gross domestic product (p < 0.001 to p = 0.002), number of healthcare establishments (p < 0.001 to p = 0.005), healthcare expenditure (p = 0.007 to p = 0.039), number of personnel in research and development (p < 0.001 to p = 0.002), and the number of neurologists (p < 0.001 to p = 0.003). The ratio of published to unpublished neurologists was negatively associated with population (p = 0.169), regional GDP (p = 0.322), number of healthcare establishments (p = 0.323), health expenditure (p = 0.155), and number of personnel in research and development (p = 0.351); however these were not statistically significant. %GDP for research and development did not show significant correlation with any of the variables (p = 0.313 to p = 0.799).”

Please see page 10, lines 4-15.

RESULTS:

Where is figure 5?

Thank you very much for bringing this to attention. Figure 5 has been appended following Section 3.7.

Please see page 10.

DISCUSSION:

Lines 21-23: How would you explain that there are more publications because of COVID?

Thank you for your inputs. An additional statement has been added to further elaborate this.

“Restructuring of the healthcare workforce and mandatory quarantine during the pandemic reduced clinical encounters and obligations2, which may have allowed neurologists more time to conduct research, particularly those that did not require direct patient interaction such as systematic reviews and/or meta-analyses and studies involving retrospective chart review. There was likely more time to complete ongoing research manuscripts. Most studies published in 2021 were case reports and case series (n=38), systematic reviews and reviews (n=20), retrospective cohort studies (n=15), and cross-sectional studies (n=13).”

Please see page 10, lines 29-36.

DISCUSSION:

Line 43: The figure 4 cited here does not reflect a correlation analysis.

Thank you for pointing this out. This has been corrected.

DISCUSSION:

Lines 67 and 71: there are random numbers here, are they perhaps references?

Thank you for noticing this. These numbers likely may have been annotation numbers for comments. These have been removed.

DISCUSSION:

Lines 72-74: expand more on the marked trend in publications about movement disorders compared with other countries in South Asia. Can you compare your numbers with those of other countries in the world as well? Do you think the Philippines has a special potential in this research topic?

Thank you for your valuable inputs.

In review of existing literature to further evaluate this and draw a comparison with other countries, we have not found similar quantification of research output for dystonia, thus comparison cannot be made unfortunately. There is indeed special potential for further research on XDP, which is being referred to in the last sentence.

DISCUSSION:

In general, the Discussion could be substantially shortened to highlight the most relevant results, without repeating information.

Thank you for your valuable inputs.

Reviewer 2 Report

The paper by Apor and Jamora presents an interesting bibliometric analysis of research productivity among Filipino neurologists, correlating it with a variety of socioeconomic factors, healthcare delivery and disease burden. As such, it is a welcomed addition to the literature within the field as it initiates important conversations about research dissemination and a variety of factors that affect it. This research included 746 publications from 1966 until December of 2021. After primary and secondary screening, which was presented in a coherent manner in the PRISM Flow Diagram, the authors were left with 245 publications.

The Introduction section is well written and informative. While it gives a general overview of the field of neurology in the Philippines, it also outlines the current socioeconomic, geographic and healthcare state within the country. A minor note within this section includes the lines 41 and 42, wherein the authors state that the Filipino neurologist is “trained to not only be a clinician but also an educator and researcher”. Since this holds true for all those that obtain an MD, especially those that proceed towards a PhD degree, and is a very vague statement, the authors should either remove this phrase (as it does not contribute to the paper) or elaborate further. Do Filipino neurologists receive additional research training that neurologists throughout the world do not? If not, there is no need to mention this.

The Materials and Methods section is elaborate and well written, including well-defined inclusion criteria. What is especially commendable here is the additional searches the authors performed to screen for all female and deceased neurologists. The method used for data collection, including accessing of the databases as well as storing and processing of data is well done. The authors used descriptive statistics to present the data. This is sufficient for this kind of analysis. Of note here, the authors should consider rephrasing the sentence in lines 202 and 203 since, as it stands now, it has dubious meaning.

The rest of the paper is well structured into logical subsections and a clear flow of ideas. The included visualizations are a welcomed addition to the current state of the art. The Discussion section is elaborate and cohesive wherein the authors put forth some very interesting observations. Of specific interest is the statement in lines 114 and 115, where the authors note the overwhelming number of research from child neurologists being published in local journals. It would be interesting here if the authors would give their opinions as to why they think this is the case.

Finally, I suggest the authors change, i.e. shorten, the title of the manuscript since it is, as it stands now, difficult to read and fully comprehend. I suggest the title to read as follows “Research productivity among Filipino neurologists associated with socioeconomic, healthcare and disease burden factors: a bibliometric analysis”. Even though the use of the English language throughout the paper is appropriate, there are many places where the authors left random numbers following sentences. These, as well as other comments relating to specific section within the paper, can be found in the attached PDF. With all that being said, I suggest the publication of this paper in IJERPH following minor revisions.

Author Response

REVIEWER #2

Comments/Recommendations

Response

The paper by Apor and Jamora presents an interesting bibliometric analysis of research productivity among Filipino neurologists, correlating it with a variety of socioeconomic factors, healthcare delivery and disease burden. As such, it is a welcomed addition to the literature within the field as it initiates important conversations about research dissemination and a variety of factors that affect it. This research included 746 publications from 1966 until December of 2021. After primary and secondary screening, which was presented in a coherent manner in the PRISMA Flow Diagram, the authors were left with 245 publications.

Thank you very much for your comments.

The Introduction section is well written and informative. While it gives a general overview of the field of neurology in the Philippines, it also outlines the current socioeconomic, geographic and healthcare state within the country. A minor note within this section includes the lines 41 and 42, wherein the authors state that the Filipino neurologist is “trained to not only be a clinician but also an educator and researcher”. Since this holds true for all those that obtain an MD, especially those that proceed towards a PhD degree, and is a very vague statement, the authors should either remove this phrase (as it does not contribute to the paper) or elaborate further. Do Filipino neurologists receive additional research training that neurologists throughout the world do not? If not, there is no need to mention this.

Thank you very much for your suggestion. The statement in the Introduction has been revised. I am also not familiar with training practices in other countries, but the PNA specifies research publication as a requirement for taking the board-certifying exam.

The Materials and Methods section is elaborate and well written, including well-defined inclusion criteria. What is especially commendable here is the additional searches the authors performed to screen for all female and deceased neurologists. The method used for data collection, including accessing of the databases as well as storing and processing of data is well done. The authors used descriptive statistics to present the data. This is sufficient for this kind of analysis. Of note here, the authors should consider rephrasing the sentence in lines 202 and 203 since, as it stands now, it has dubious meaning.

Thank you very much for your comments. We would like to verify if lines 202 and 203 being referred to under Materials and Methods pertains to lines 102 and 103 regarding the socioeconomic parameters? If this is the case, this has been further specified as follows:

“Region-specific socioeconomic parameters (gross domestic product (GDP), population, health expenditure, number of health establishments, %GDP allocated for research and development (R&D), and number of R&D personnel) were sourced from official government reports under the Philippine Statistics Authority [21,22] and the Compendium of Science and Technology [23]. Health expenditure is determined from health financing revenues using the Philippine National Health Accounts framework. Research expenditure is based on the %GDP allocated for R&D.”

Please see page 3, lines 102-104.

The rest of the paper is well structured into logical subsections and a clear flow of ideas. The included visualizations are a welcomed addition to the current state of the art. The Discussion section is elaborate and cohesive wherein the authors put forth some very interesting observations. Of specific interest is the statement in lines 114 and 115, where the authors note the overwhelming number of research from child neurologists being published in local journals. It would be interesting here if the authors would give their opinions as to why they think this is the case.

Thank you very much for your insights.

Given the generally lower counts of child neurology research studies, it is difficult to draw conclusions, however our thoughts have been appended as an additional statement:

“It may be of note that all three pediatric neurology training institutions have their own institutional local publication, which may pose as more accessible and affordable options for research dissemination.”

Please see page 12, lines 131-133.

Finally, I suggest the authors change, i. e. shorten, the title of the manuscript since it is, as it stands now, difficult to read and fully comprehend. I suggest the title to read as follows “Research productivity among Filipino neurologists associated with socioeconomic, healthcare and disease burden factors: a bibliometric analysis”. Even though the use of the English language throughout the paper is appropriate, there are many places where the authors left random numbers following sentences. These, as well as other comments relating to specific section within the paper, can be found in the attached PDF. With all that being said, I suggest the publication of this paper in IJERPH following minor revisions.

Thank you very much for your suggestion.

The random numbers have been removed. We have also revised the title as suggested to be more succinct.

Round 2

Reviewer 1 Report

Thank you for considering my recommendations.
I advise acceptance of this paper in its current version.